# Diaporine Potentiates the Anticancer Effects of Oxaliplatin and Doxorubicin on Liver Cancer Cells

**DOI:** 10.3390/jpm12081318

**Published:** 2022-08-16

**Authors:** Shiliu Tian, Rui Su, Ke Wu, Xuhan Zhou, Jaydutt V. Vadgama, Yong Wu

**Affiliations:** 1Key Laboratory of Exercise and Health Sciences of Ministry of Education, Shanghai University of Sport, Shanghai 200438, China; tianshiliu@sus.edu.cn; 2Fujian Sports Vocational Education and Technical College, Fuzhou 350001, China; 3College of Engineering, University of California, Berkeley, CA 94720, USA; rui999@berkeley.edu; 4David Geffen UCLA School of Medicine and UCLA Jonsson Comprehensive Cancer Center, Division of Cancer Research and Training, Department of Internal Medicine, Charles Drew University of Medicine and Science, Los Angeles, CA 90095, USA; kewu@cdrewu.edu (K.W.); jayvadgama@cdrewu.edu (J.V.V.); 5Fulgent Life Inc., Irvine, CA 92620, USA; jameszhou201801@gmail.com

**Keywords:** liver cancer, diaporine, oxaliplatin, doxorubicin, cell proliferation, apoptosis

## Abstract

Recent studies have shown that diaporine, a novel fungal metabolic product, has a strong in vitro and in vivo anticancer effect on human non-small-cell lung and breast cancers. In this study, three human hepatocarcinoma cell lines (HepG2, Hep3B, and Huh7) were used to evaluate the efficacy of diaporine alone and in combination with the standard cytotoxic drugs oxaliplatin and doxorubicin for the treatment of liver cancer. We demonstrated that diaporine, oxaliplatin, and doxorubicin triggered a concentration- and time-dependent decrease in the number of HepG2 cells. Diaporine at a concentration of 2.5 μM showed almost 100% inhibition of cell counts at 72 h. Similar effects were observed only with much higher concentrations (100 μM) of oxaliplatin or doxorubicin. Decreases in cell numbers after 48 h treatment with diaporine, oxaliplatin, and doxorubicin were also demonstrated in two additional hepatoma cell lines, Hep3B and Huh7. The combination of these drugs at low concentration for 48 h in vitro noticeably showed that diaporine improved the inhibitory effect on the number of cancer cells induced by oxaliplatin or doxorubicin. Additionally, this combination effectively inhibited colony growth in vitro. We found that inhibition of phosphorylation of ERK1/2 significantly increased when diaporine was used in combination with other agents. In addition, we also found that when diaporine was used in combination with doxorubicin or oxaliplatin, their proapoptotic effect greatly increased. We further revealed that the induction of apoptosis in hepatoma cells after treatment is due, at least in part, to the inhibition of phosphorylation of AKT, leading to the activation of caspase-3, inactivation of poly (ADP-ribose) polymerase (PARP), and subsequently to DNA damage, as indicated by the increased level of H2AX. Based on these findings, we suggest that diaporine in combination with the standard cytotoxic drugs oxaliplatin and doxorubicin may play a role in the treatment of liver cancer.

## 1. Introduction

Cancer is a devastating disease. Regardless of the development of disease diagnosis, treatment, and preventive measures, it is one of the main causes of death and morbidity in the world [1,2,3]. Moreover, the number of cases is increasing, which is estimated to reach 21 million by 2030 [4]. Hepatocellular carcinoma (HCC) is a kind of cancer that originates from the liver and is an invasive tumor that often occurs in the context of chronic liver disease and cirrhosis. Primary liver cancer is the fifth most common cancer in men, the seventh most common cancer in women, and the third largest cancer-related death cause in the world [5,6]. The incidence of liver cancer in the USA is rising, reaching 4.5/100,000 per year in 2005 [7]. Chemotherapy is one of the most imperative treatments for advanced liver cancer. In the past decade, the survival rate of patients with metastatic liver cancer has increased with the application of targeted drugs such as sorafenib, nivolumab, and cabozantinib [8]. Presently, several new joint programs including these drugs are being developed. Although the treatment of liver cancer has progressed, it is still one of the most stubborn cancers to treat. The recurrence of liver cancer is still the main problem after radical treatment, and the incidence rate is over 70% after 5 years [9].

At present, cancer treatment methods mainly include tumor surgery, radiotherapy, immunotherapy, chemotherapy, cancer vaccination, photodynamic therapy, stem cell transformation, or their combination. These methods are often accompanied by serious side effects. The shortcomings of these treatments include damage to normal tissues, limited bioavailability, toxicity, nonspecificity, quick clearance, etc. [10,11]. Plants are repositories of new chemicals, which provide a promising route for cancer research. For decades, patients, oncologists, and scientists have expressed interest in employing natural compound drugs because the overall results and side effects of current cytotoxic drugs and targeted therapy are disappointing. Natural products provide abundant resources for developing new anticancer drugs. In particular, it was reported that over 50% of modern anticancer drugs originated from natural products [12], and natural products will continue to be the source of new lead compounds [13]. Endophytic fungi in host plants are abundant sources of bioactive natural products [14,15]. Diaporine is an extraordinary symmetric polyketone compound extracted from endophytic fungi, and its structure was explored via broad spectroscopic analysis [16]. In previous studies, it has been found that diaporine is not only a modulator of macrophage differentiation [16] but also hinders the growth of non-small-cell lung cancer (NSCLC) by regulating the miR-99a/mTOR signaling pathway [17]. Nonetheless, there are no reports about the toxicological characteristics of this natural product and its killing effect on liver cancer cells.

We know from clinical trials and clinical practice that single-drug therapy seldom brings clinical benefits to cancer patients, while combined therapy is essential for the effective treatment of tumors. In this study, we investigated the potential anticancer effects of diaporine alone and in combination with standard cytotoxic drugs oxaliplatin and doxorubicin in the treatment of liver cancer.

## 2. Materials and Methods

### 2.1. Cell Culture and Reagents

The human liver cancer cell line HepG2 was originally obtained from the American Type Culture Collection (Manassas, VA, USA). Hep3B and Huh7 cells were from the Cell Bank of the Chinese Academy of Sciences (Shanghai, China). The cells were maintained in Dulbecco’s modified Eagle medium (DMEM) (Gibco, Shanghai, China) supplemented with 10% fetal bovine serum (FBS), nonessential amino acids, and antibiotics. In all experiments using the trypan blue dye exclusion method, the cell survival rate was higher than 99%. Oxaliplatin and doxorubicin were purchased from Sigma-Aldrich (Saint Louis, MO, USA). Phospho-AKT and phospho-ERK1/2 antibodies were purchased from Cell Signaling Technology (Danvers, MA, USA). Cleaved caspase-3 (Asp175) and cleaved poly (ADP-ribose) polymerase (PARP) were purchased from Bioworld (Nanjing, Jiangsu, China), AKT antibody from Abcam, phospho-histone H2AX antibody from Millipore, and the ERK2 and β-actin antibodies from Santa Cruz Biotechnology, Inc. (Dallas, TX, USA). Diaporine was extracted from endophytic fungi; prior to each experiment, it was dissolved in dimethyl sulfoxide (DMSO).

### 2.2. Effect of Diaporine, Oxaliplatin, Doxorubicin and Their Combination on the Number of Liver Cancer Cells

Cells were seeded into 6-well plates at a density of 50,000 cells per well. After 24 h, the cells were treated with 0.1–5 µM diaporine, 1–100 µM oxaliplatin, or 1–100 µM doxorubicin for another 24, 48, and 72 h. The control cultures were incubated with 0.1% DMSO (vehicle). In another set of experiments, cells were treated with a combination of diaporine/oxaliplatin or diaporine/doxorubicin. The effect of these combinations on the number of cells was measured at the specified time using Scepter 2.0 Handheld Automated Cell Counter (Millipore). By comparing drug-treated with DMSO-treated cells (the number of cells was assumed to be 100%), the data are expressed as a proportional number of cells (%).

### 2.3. Analysis of Drug Interactions

To quantify drug interactions between diaporine and oxaliplatin or doxorubicin, CompuSyn software was used. All simulations were conducted assuming that the two drugs were combined in a nonfixed ratio of doses with a fixed concentration of diaporine (Drug A) and variable concentrations of oxaliplatin or doxorubicin (Drug B). CompuSyn (ComboSyn, Inc., New York, NY, USA) was employed to determine the combination index (CI), a quantitative representation of pharmacological interactions. This reference model is based on the unified theory introduced by Chou and Talalay [18]. The CI was plotted on the Y axis as a function of the effect level (Fa) on the X axis to evaluate the drug synergism between drug combinations. CI < 1 indicates a synergistic effect, CI = 1 indicates an additive effect, and CI > 1 indicates an antagonistic effect.

### 2.4. Effect of Different Treatments on Colony Growth in Matrigel Matrix

A layer of Matrigel (150 µL) was transferred to the wells of a 24-well cell culture dish and allowed to stand at 37 °C for 30 min. A second layer (300 µL), consisting of 150 µL of Matrigel dissolved in a growth medium (150 µL) containing 1.5 × 10^3^ cells, was placed on top of the first layer in a humidified incubator at 37 °C for 30 min. About 0.5 mL of growth medium was added to the top of the second layer. The cells were then cultured in a humidified incubator at 37 °C for 14 days and then treated with diaporine, oxaliplatin, doxorubicin, or a combination thereof for another 7 days. The control cells were treated with 0.1% DMSO. The medium was refreshed two times per week. At the end of the experiment, the colonies were stained with Giemsa (2%) for 1 h and incubated overnight with PBS to eliminate extra Giemsa staining. The colonies were photographed and scored, and the percentage of colonies greater than 100 µm was measured. The data obtained were used to compare drug-treated with DMSO-treated colonies. Colonies greater than 100 µm are expressed as a percentage of the total number of colonies counted and were compared with the DMSO-treated control groups.

### 2.5. Quantification of Apoptosis

Apoptosis was determined using Annexin V/7-aminoactinomycin D (7-AAD) double-staining assay according to the manufacturer’s instructions of the Muse^®^ Annexin V & Dead Cell Kit (MCH100105, EMD Millipore, Billerica, MA, USA). Briefly, the HepG2 cells were incubated with or without each drug alone or in combination for 48 h. Cells were collected and stained with Annexin V-FITC and 7-AAD according to the Guava^®^ Muse^®^ Cell Analyzer protocol. Fluorescence intensity was measured by flow cytometry using a Muse™ Cell Analyzer (EMD Millipore), and the cells were sorted into live (annexin V-/7-AAD-), early apoptotic (annexin V+/7-AAD-), late apoptotic (annexin V+/7-AAD+), and necrotic (annexin V-/7-AAD+) cells. The sum of the early and late apoptotic cells was calculated as the total number of apoptotic cells.

### 2.6. Effects of Diaporine, Oxaliplatin, Doxorubicin, and Their Combination on Expression and Phosphorylation of Proapoptotic and Antiproliferative Proteins

The cells were inoculated in a 100 mm petri dish with 2 × 10^6^ cells per dish for 24 h, then treated with diaporine (0.5 µM), oxaliplatin (10 µM), doxorubicin (10 µM), or a combination of diaporine/oxaliplatin or diaporine/doxorubicin for another 24 h. Control groups were incubated with 0.1% DMSO (vehicle). In the second group of experiments, the cells were exposed to an increasing concentration of diaporine (0.5–2.5 µM) for 2 h. The total proteins of the vehicle- and drug-treated cells were isolated using a standard RIPA buffer. The cell lysate was recovered through centrifuging at 14,000 rpm for 20 min at 4 °C to eliminate the insoluble substance, and the protein concentration of the lysate was measured by a BCA protein assay kit (Thermo Fisher Scientific, Waltham, MA, USA). Proteins (20 µg) were separated by SDS-PAGE gel to detect the expression and phosphorylation levels of different proapoptotic and antiproliferative proteins. After electrophoresis, the proteins were transferred to nitrocellulose (NC) membranes, blocked with 5% nonfat milk at room temperature for 2 h, and incubated overnight at 4 °C with specific primary antibodies and β-actin (1:1000). The blots were washed, incubated with the corresponding secondary antibodies, and visualized with the ECL system (Thermo Fisher Scientific, Waltham, MA, USA). Membrane stripping was carried out via incubating the membrane in stripping buffer (Thermo Fisher Scientific) based on the manufacturer’s instructions. Densitometry analysis was carried out with an HP Deskjet F4180 Scanner (HP Development Company, Palo Alto, CA, USA) and ImageJ software.

### 2.7. Statistics

The results are expressed as the means ± SEM of the data presented. Differences between test and control values were calculated with ANOVA followed by Dunnett’s post hoc multiple comparison test. *** *p* < 0.001, ** *p* < 0.01, and * *p* < 0.05 indicate that the difference was statistically significant.

## 3. Results

### 3.1. Effect of Diaporine, Oxaliplatin, and Doxorubicin on Cell Numbers

As shown in Figure 1, diaporine (0.1–5 μM), oxaliplatin (1–100 μM), and doxorubicin (1–100 μM) reduced the number of HepG2 cells in a concentration- and time-dependent manner within 24, 48, and 72 h (Figure 1A–C). The dose-dependent effects of diaporine, oxaliplatin, and doxorubicin were also confirmed on two other liver cancer cell lines, Hep3B and Huh7 (Figure 1D–F). The IC50 concentration (48 h) of diaporine in the HepG2 cells was 0.5 μM and about 0.76 μM in both Hep3B and Huh7 cells (Figure 1G). The IC50 of oxaliplatin and doxorubicin was much higher than that of diaporine (Figure 1G).

### 3.2. Diaporine Improved the Antitumor Activity of Oxaliplatin and Doxorubicin against HEPG2 Cells

Treatment of the HepG2 cells with diaporine (0.5 μM) at the IC50 concentration for 48 h significantly enhanced the inhibitory effect of oxaliplatin (1–10 μM) (Figure 2A) and doxorubicin (0.5–2.5 μM) (Figure 2B) on cell growth. To quantify the drug interactions between diaporine and oxaliplatin or doxorubicin, we estimated the combination index (CI) using CompuSyn software. The CI values of the diaporine/oxaliplatin combination were lower than one (Figure 2C), suggesting that the combination of these compounds has a synergistic effect on the growth inhibition of HepG2 cells. The same behavior was seen for combinations involving diaporine and doxorubicin (Figure 2D). The results obtained using CompuSyn software are summarized in Table 1.

Next, we studied the effects of these combinations on the growth of the HEPG2 colonies that had formed. First, the HepG2 cells were allowed to grow and form noticeable colonies without any treatment. After 2 weeks of culture, with DMSO as a control, diaporine (0.5 μM), oxaliplatin (10 μM), doxorubicin (10 μM) or their combination was added to the colonies for 1 week. While the number of colonies acquired in each treatment did not seem to change, we noticed that the size of the colonies significantly decreased. Although large colonies accounted for about 70% of the total number of colonies in the control, they only accounted for 25% and 35% of the total number of colonies treated with oxaliplatin (Figure 3A) and doxorubicin (Figure 3B) alone, respectively. Compared with oxaliplatin or doxorubicin alone, diaporine did not cause significant colony growth inhibition, but when used in combination, it significantly enhanced oxaliplatin (Figure 3A) or doxorubicin-mediated colony growth inhibition (Figure 3B). We speculate that the effects of these drugs on colony growth when used alone or in combination may be caused by cell death or inhibition of cell proliferation.

The MAPK signaling pathway is primarily involved in regulating cell proliferation, and blocking this pathway can inhibit the growth of liver tumors [19]. ERK1 and ERK2 are the ultimate effectors of the MAPK pathway. They are activated by phosphorylation, which leads to the activation of various substrates, thus inducing cell proliferation. Therefore, we decided to study the activation of ERK1/2 when diaporine was used alone or in combination with oxaliplatin and doxorubicin. Intriguingly, although we observed that a single drug had an obvious inhibitory effect on ERK1/2 phosphorylation, the combined drugs enhanced this inhibitory effect (Figure 3C).

### 3.3. Diaporine Combined with Doxorubicin or Oxaliplatin Promotes Apoptosis of HepG2 Hepatoma Cells

Next, we studied whether the further decrease in HepG2 cell number resulted from the increase in apoptotic cell death when diaporine was used in combination with doxorubicin or oxaliplatin. For this purpose, the HepG2 liver cancer cells were treated with diaporine (0.5 µM) alone or in combination with oxaliplatin (10 μM) or doxorubicin (10 μM) for 48 h. The apoptotic rate was detected by Annexin V/7-AAD staining, as illustrated in Figure 4. The combination of diaporine and oxaliplatin increased the level of apoptosis compared with either drug alone. In addition, we observed that the combination of diaporine and doxorubicin resulted in more apoptosis. The number of apoptotic cells increased from 22% when using diaporine (0.5 µM) alone and 24% when using doxorubicin (10 µM) alone to 66% when used in combination, indicating that, at these concentrations, there may be a synergy between diaporine and doxorubicin (Figure 5).

### 3.4. Effects of Diaporine, Oxaliplatin, and Doxorubicin Alone or in Combination on Survival, Apoptosis Pathway, and DNA Damage

When the three drugs were used alone, they all significantly inhibited the phosphorylation of Akt. Nevertheless, no further decreases in phosphorylation levels were observed with the combination of the three drugs (Figure 6A). As expected, none of these treatments affected the level of AKT total protein (Figure 6A). Previous studies have suggested that downregulation of AKT phosphorylation can induce caspase-3-dependent apoptosis [20,21]. Consistent with these reports, oxaliplatin-induced dephosphorylation of Akt resulted in the accumulation of caspase-3 cleavage [22]. We previously found that diaporine induces caspase-3 cleavage in breast cancer cells (unpublished data). In this context, the apoptosis induced by diaporine alone and in combination with oxaliplatin and doxorubicin was further evaluated by detecting caspase-3 activation and subsequent cleavage and inactivation of PARP. Treatment with diphorine (0.5 μM), oxaliplatin (10 μM), and doxorubicin (10 μM) for 24 h activated caspase-3, a crucial step in inducing apoptosis. When diaporine was combined with oxaliplatin or doxorubicin, this activation was further marginally enhanced (Figure 6A). Activation of caspase-3 resulted in cleavage and inactivation of downstream PARP, a nuclear protein implicated in DNA repair and apoptosis. Similarly, downstream PARP cleavage was also detected in the HEPG2 cells treated with single and combined drugs.

Inhibition of PARP with PARP inhibitors is known to enhance the level of H2AX phosphorylation associated with DNA damage [23]. Phosphorylated H2AX histone (γH2AX) is a very important marker of DNA double strand break. It has been reported that doxorubicin can induce DNA damage in liver cancer cells [24,25]. To verify whether the cell death induced by diaporine, oxaliplatin, and doxorubicin is related to DNA damage, these drugs were applied, individually or in combination, to HepG2 cells, and the expression of γ-H2AX in the total protein was detected. We observed that cells treated with the three drugs alone suffered DNA damage. Remarkably, the combination of these drugs further increased the level of DNA damage in the treated cells (Figure 6A). To ascertain if diaporine mediates its effects by DNA damage, inhibition of the ERK1/2 and Akt signaling pathway, or a combination of both, we carried out an experiment to detect DNA damage, p-Akt, and p-ERK1/2 levels after treatment with diaporine (0.5–2.5 µM) for 2 h. Figure 6B shows that the 0.5 µM diaporine used in this study had no effect on DNA damage, but it significantly mitigated the levels of phosphorylated AKT and ERK. The DNA damage occurred at very low levels only at very high doses of diaporine. Therefore, this result suggests that the inhibition of ERK/Akt signaling was an early event in response to diaporine treatment and that DNA damage was a late event, possibly due to excessive DNA damage caused by active apoptosis. Our results demonstrate that the combined use of diaporine promotes the antiproliferative and proapoptotic effects of oxaliplatin and doxorubicin.

## 4. Discussion

Diaporine is a new fungal metabolite with an unprecedented symmetrical polyketone structure [16]. This study found, for the first time, that diaporine treatment can significantly impede the proliferation of liver cancer cells and induce their apoptosis. When diaporine is used in combination with DNA-damaging agents oxaliplatin or doxorubicin, the effect of inhibiting HepG2 cell proliferation and inducing apoptosis, thus inhibiting the growth of HepG2 colony, is better than that of using any cytotoxic drug alone.

Our results show that the inactivation of proliferation (ERK) and survival-promoting (Akt) pathways, the activation of caspase-3, cleavage of PARP, and resultant DNA damage at least partially explain the anticancer effect of diaporine. In addition, combination treatment with oxaliplatin or doxorubicin promoted these effects, further elucidating the ability of diaporine to augment the apoptosis mainly induced by doxorubicin. The data also show that the inhibition of ERK1/2 phosphorylation by single therapy and its elevated suppression under combined conditions indicate a potential effect of the treatments on the proliferation of hepatocellular carcinoma cells.

The double inhibitory effect of our treatment on the PI3K/AKT and Raf/MEK/ERK pathways is consistent with the interaction between these two pathways previously reported [26,27,28,29,30,31,32]. Here, we prove that inhibiting phosphorylation of AKT and ERK is the key to the antitumor activity of diaporine. These results are consistent with those of previous studies indicating that bioactive substances derived from fungi target signaling machinery such as the PI3K/Akt signaling pathway, etc., in various cancer cells [33,34,35,36].

A previous study [37] assessed the sensitivity of diaporine to several cancer cells and showed that diaporine can inhibit the proliferation of cancer cells, especially breast cancer cells. Furthermore, it can impair the morphology and viability of breast cancer cells and also damage their migration and invasion ability. Further research showed that diaporine can effectively prompt strong cell cycle arrest in breast cancer cells, especially in the G2 phase, which eventually leads to the apoptosis of cancer cells. The mechanism of diaporine mediating breast cancer cell death is related to its ability to induce the increase in ROS levels in these cells.

This study provides a theoretical basis for in vivo studies to substantiate the relevance of combined therapy. We think that diaporine combined with standard cytotoxic drugs such as oxaliplatin and doxorubicin can improve the therapeutic effect of liver cancer treatment. Plant chemistry is a research field with broad prospects. The active natural product compound diaporine described in this study should be further studied in clinical trials to evaluate their possible application, toxicology, and special genotoxicity characteristics for various cancers in vitro or in vivo.

## Figures and Tables

**Figure 1 jpm-12-01318-f001:**
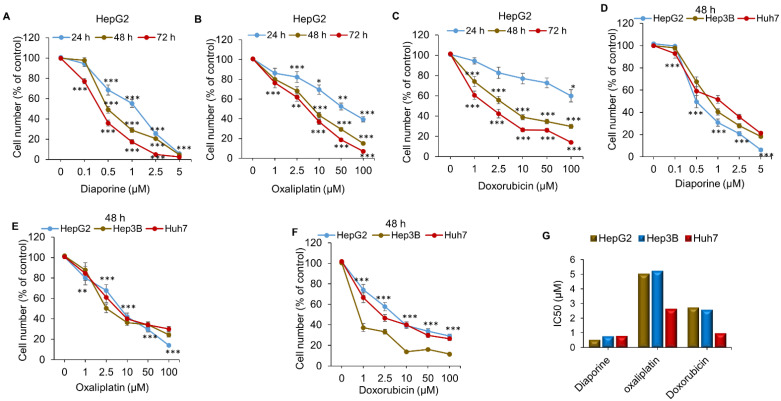
Comparison of the inhibitory effect of diaporine on the number of hepatocellular carcinoma cells with oxaliplatin and doxorubicin. Exponentially growing HepG2 cells were incubated with vehicle (0.1%DMSO), diaporine (**A**), oxaliplatin (**B**), and doxorubicin (**C**) at the indicated concentration for 24–72 h. The effects of these three drugs on the other two hepatoma cell lines, Hep3B and HuH7, were confirmed at 48 h (**D**–**F**). Cell counts were performed as described in the Materials and Methods. (**G**). IC50 (μM) of diaporine compared with those of oxaliplatin and doxorubicin at 48 h of treatment. All the experiments were independently repeated 3 times. The data are presented as SEM. * *p* < 0.05; ** *p* < 0.01; *** *p* < 0.001.

**Figure 2 jpm-12-01318-f002:**
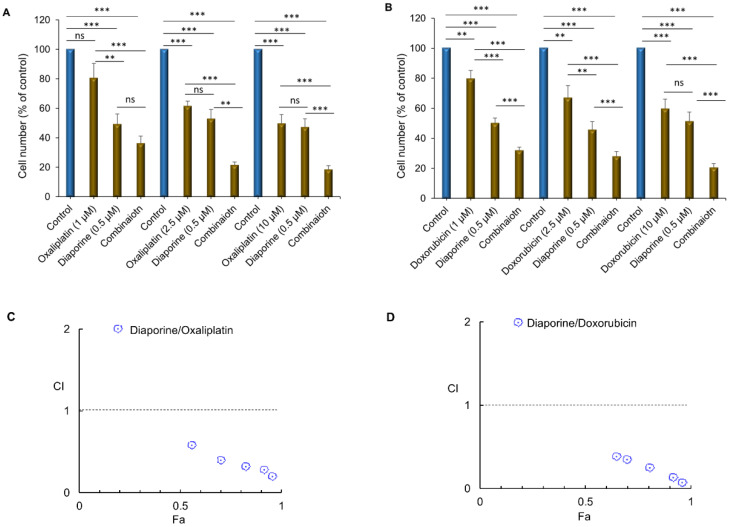
Diaporine augments the inhibition of HepG2 cell numbers by (**A**) oxaliplatin (1–10 μM) and (**B**) Doxorubicin (0.5–2.5 μM). Cells were treated with the indicated drugs for 48 h, and all experiments were repeated at least 3 times. The data are presented as SEM. ** *p* < 0.01; *** *p* < 0.001; ns, not significant. Diaporine has synergic cytotoxic effects with oxaliplatin (**C**) or doxorubicin (**D**) in HepG2 cells. Combination index (CI) plots (Fa-CI plot) were generated by CompuSyn software (ComboSyn Inc., Paramus, NJ, USA) according to the Chou–Talalay method. Synergy is present when CI < 1.0 Fa: inhibitory effect, CI: combination index.

**Figure 3 jpm-12-01318-f003:**
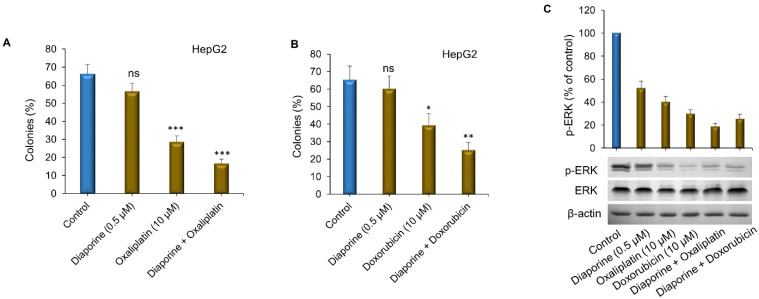
Diaporine enhances the inhibitory effects of (**A**) oxaliplatin (10 µM) and (**B**) doxorubicin (10 µM) on colony growth. The data are displayed as the mean percentage histogram of macrocolony growth ± SEM. (**C**) Diaporine can enhance the inhibition of oxaliplatin and doxorubicin on ERK phosphorylation in HepG2 hepatoma cells. Densitometry analysis of p-ERK from 3 independent experiments (upper panel). Β-actin was used as an internal control at protein level. * *p* < 0.05; ** *p* < 0.01; *** *p* < 0.001; ns, not significant.

**Figure 4 jpm-12-01318-f004:**
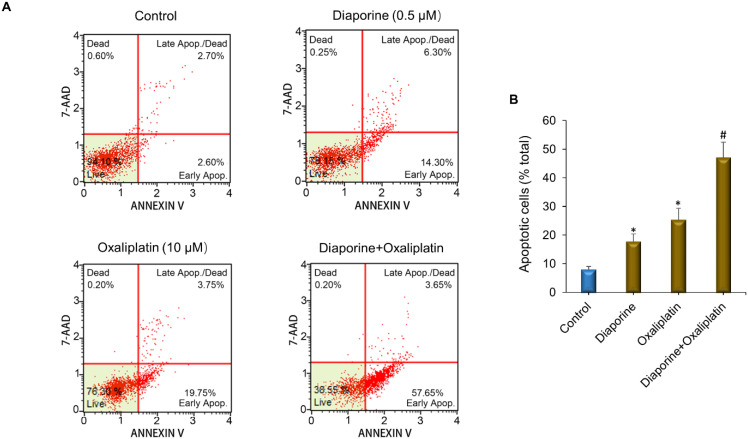
Effect of diaporine on apoptosis of HEPG2 hepatoma cells induced by oxaliplatin. (**A**) HepG2 cells were treated separately or jointly with diaporine (0.5 μM) and oxaliplatin (10 μM) for 48 h. The cells were stained with Annexin V/7-aminoactinomycin D (7-AAD) according to Guava^®^ Muse^®^ Cell Analyzer protocol. (**B**) Data are presented as the mean ± SEM. of 3 independent experiments. * *p* < 0.05 vs. control; # *p* < 0.05 vs. oxaliplatin.

**Figure 5 jpm-12-01318-f005:**
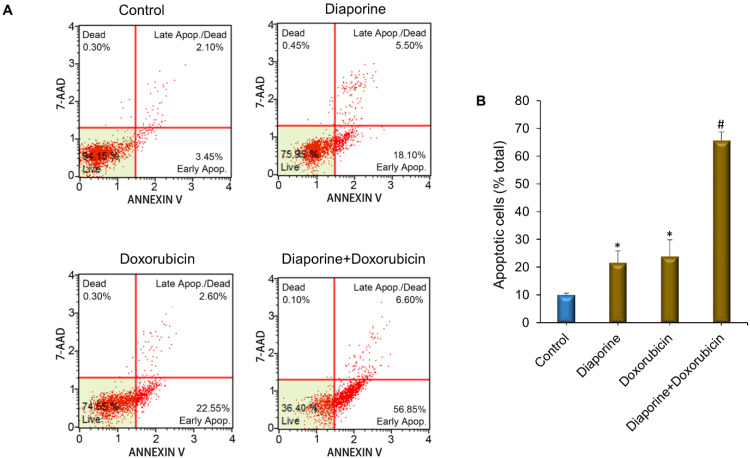
Diaporine augments apoptotic cell death induced by doxorubicin in HepG2 liver cancer cells. (**A**) HepG2 cells were incubated with diaporine (0.5 μM) and doxorubicin (10 μM) separately and in combination for 48 h. The cells were stained with Annexin V/7-aminoactinomycin D (7-AAD) according to Guava^®^ Muse^®^ Cell Analyzer protocol. (**B**) Data are presented as the mean ± SEM of 3 independent experiments. * *p* < 0.05 vs. Control; # *p* < 0.05 vs. doxorubicin.

**Figure 6 jpm-12-01318-f006:**
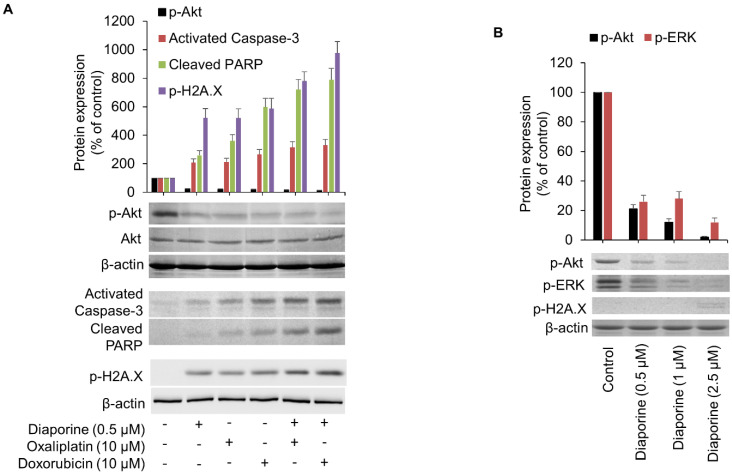
Effects of diaporine, oxaliplatin, doxorubicin, or their combinations on the AKT phosphorylation, caspase-3 activation, cleaved Poly (ADP-ribose) polymerase (PARP), and H2AX phosphorylation. (**A**) HepG2 cells were incubated with the indicated concentrations of diaporine, oxaliplatin, doxorubicin, or their combinations for 24 h. The whole lysates were analyzed using Western blotting with indicated antibodies. Densitometry analysis was from 3 independent experiments (upper panel). Compared with the control sample, which was considered to be equal to 100%, the normalized band density is expressed as a percentage change. (**B**) Changes in p-AKT, p-ERK and p-H2AX levels after diaporine treatment (0.5–2.5 µM) for 2 h.

**Table 1 jpm-12-01318-t001:** The nature of drug interactions in HepG2 cells treated with diaporine and combined with oxaliplatin or doxorubicin.

Drug A	Dose A (µM)	Drug B	Dose-B (µM)	Effect (Fa)	CI Value	Interaction
Diaporine	0.5	Oxaliplatin	1.0	0.5624	0.56	Synergism
			2.5	0.7032	0.39	Synergism
			10.0	0.8211	0.31	Synergism
			50.0	0.9134	0.26	Synergism
			100.0	0.9512	0.18	Synergism
Diaporine	0.5	Doxorubicin	1.0	0.6512	0.39	Synergism
			2.5	0.7018	0.35	Synergism
			10.0	0.8011	0.25	Synergism
			50.0	0.9023	0.17	Synergism
			100.0	0.9612	0.06	Synergism

## Data Availability

Data are contained within the article.

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
