# Peer review of "Diaporine Potentiates the Anticancer Effects of Oxaliplatin and Doxorubicin on Liver Cancer Cells"

_jpm, 2022, doi:10.3390/jpm12081318_

Round 1

Reviewer 1 Report

It was my pleasure to review this manuscript, which explore the effect of diaporine in combination with oxaliplatin or doxorubicin in liver cancer cells. My comments are as follows:

1. The authors revealed the effect of diaporine in combination with oxaliplatin or doxorubicin in liver cancer cells. How about trying test and discuss whether these effects are additive or synergistic with reference to following study.

  Ting-Chao Chou. Theoretical basis, experimental design, and computerized simulation of synergism and antagonism in drug combination studies. Pharmacol Rev. 2006, 58(3):621-81.

2. The initial letters of drug names, for example oxaliplatin, doxrubicin and are capitalized or lowercased in the text and are not consistent. 

Author Response

Manuscript number: jpm-1834604

Title: Diaporine Potentiates the Anti-Cancer Effects of Oxaliplatin and Doxorubicin on liver cancer Cells

We would like to thank the editors and reviewers for your careful readings and for the thoughtful comments and constructive suggestions, which guided our revisions resulting in this improved paper. We are very delighted to know that the reviewers consider "The manuscript is well structured, very well written and is novel contribution toward this journal." Thanks to these very positive comments, we are extremely encouraged to revise the manuscript according to reviewers' comments. Each comment has been carefully considered point by point and responded. We are confident that the new version of the manuscript will be greatly improved. We respond below in detail to each of the reviewer’s comments and also highlight the changes to the manuscript within the document by using the “Track Changes” function.

Reviewer 1

Comments and Suggestions for Authors

It was my pleasure to review this manuscript, which explore the effect of diaporine in combination with oxaliplatin or doxorubicin in liver cancer cells. My comments are as follows:

  1. The authors revealed the effect of diaporine in combination with oxaliplatin or doxorubicin in liver cancer cells. How about trying test and discuss whether these effects are additive or synergistic with reference to following study.

  Ting-Chao Chou. Theoretical basis, experimental design, and computerized simulation of synergism and antagonism in drug combination studies. Pharmacol Rev. 2006, 58(3):621-81.

Response: We appreciate your comments very much. As you suggested, we have added analysis of drug interactions to quantify drug interactions between diaporine and oxaliplatin or doxorubicin. We estimated the combination index (CI) by using CompuSyn software in our revised manuscript [marked] (Page 5, line 192; Figure 2C and 2D; Table 1).

  1. The initial letters of drug names, for example oxaliplatin, doxrubicin and are capitalized or lowercased in the text and are not consistent. 

Response: We greatly appreciate the reviewer’s efforts to carefully review the paper and the valuable suggestions offered. As suggested by the reviewer, we changed the name of the drug in the manuscript to lowercase.

Reviewer 2 Report

Dear Editor,

The manuscript “title- Diaporine Potentiates the Anti-cancer Effects of Oxaliplatin and Doxorubicin on Liver Cancer Cells” is well structured, very well written and is novel contribution toward this journal. In my view paper is suitable for publication with following minor revisions

The authors have not discussed about cancer, please write 1-2 lines about cancer as a prevailing disease.

It would be better to add data about Worldwide Cancer statistics. Cite and Support your introduction by reading the articles links provided below.

Different treatment strategies for cancer treatment like chemo-, radio-, and phyto therapies. Read the articles for different treatment strategies.  The articles links are provided below to support your introduction.

1-      https://pubmed.ncbi.nlm.nih.gov/29535002/

2-      https://pubmed.ncbi.nlm.nih.gov/30551389/

Please also discuss why author have used/ selected Diaporine as potentiate?

Please also add 1-2 line of future prospective. What can be done next and what are the gaps that can be filled?

Author Response

Manuscript number: jpm-1834604

Title: Diaporine Potentiates the Anti-Cancer Effects of Oxaliplatin and Doxorubicin on liver cancer Cells

We would like to thank the editors and reviewers for your careful readings and for the thoughtful comments and constructive suggestions, which guided our revisions resulting in this improved paper. We are very delighted to know that the reviewers consider "The manuscript is well structured, very well written and is novel contribution toward this journal." Thanks to these very positive comments, we are extremely encouraged to revise the manuscript according to reviewers' comments. Each comment has been carefully considered point by point and responded. We are confident that the new version of the manuscript will be greatly improved. We respond below in detail to each of the reviewer’s comments and also highlight the changes to the manuscript within the document by using the “Track Changes” function.

Reviewer 2

Dear Editor,

The manuscript “title- Diaporine Potentiates the Anti-cancer Effects of Oxaliplatin and Doxorubicin on Liver Cancer Cells” is well structured, very well written and is novel contribution toward this journal. In my view paper is suitable for publication with following minor revisions

The authors have not discussed about cancer, please write 1-2 lines about cancer as a prevailing disease.

Response: We greatly appreciate the reviewer’s efforts to carefully review the paper and the valuable suggestions offered. As suggested by the reviewer, we have added 1-2 lines about cancer as a prevailing disease to the Introduction section of our revised manuscript [marked] (Page 1, line 40).

It would be better to add data about Worldwide Cancer statistics. Cite and Support your introduction by reading the articles links provided below.

Different treatment strategies for cancer treatment like chemo-, radio-, and phyto therapies. Read the articles for different treatment strategies.  The articles links are provided below to support your introduction.

  • https://www.sciencedirect.com/science/article/pii/S2221169117308730
  • https://pubmed.ncbi.nlm.nih.gov/29535002/
  • https://pubmed.ncbi.nlm.nih.gov/30551389/
  • https://www.apjtm.org/article.asp?issn=19957645;year=2018;volume=11;issue=9;spage=501;epage=509;aulast=Abbasi

Please also discuss why author have used/ selected Diaporine as potentiate?

Response: We appreciate your comments very much. As you suggested, we have added relevant information to our revised manuscript [marked] (Page 2, line 55).

Please also add 1-2 line of future prospective. What can be done next and what are the gaps that can be filled?

Response: We have added relevant information to our revised manuscript [marked] (Page 10, line 323).